# Sensor for the Characterization of 2D Angular Actuators with Picoradian Resolution and Nanoradian Accuracy with Microradian Range

**DOI:** 10.3390/s20247034

**Published:** 2020-12-09

**Authors:** Marco Pisani, Milena Astrua, Srijith Bangaru Thirumalai Raj

**Affiliations:** 1Istituto Nazionale di Ricerca Metrologica, INRIM, 10135 Torino, Italy; m.astrua@inrim.it (M.A.); s.thirumalai@inrim.it (S.B.T.R.); 2Politecnico di Torino, 10129 Torino, Italy

**Keywords:** angular actuators, angular sensors, encoders, autocollimators

## Abstract

High precision angular actuators are used for high demanding applications such as laser steering for photolithography. Piezo technology allows developing actuators with a resolution as low as a few nanoradians, with bandwidths as high as several kilohertz. In most demanding applications, the actual performance of these instruments needs to be characterized. The best angular measurement instruments available today do not sufficient resolution and/or bandwidth to satisfy these needs. At the Istituto Nazionale di Ricerca Metrologica, INRIM a device was designed and built aiming at characterizing precision 2D angular actuators with a resolution surpassing the best devices on the market. The device is based on a multi reflection scheme that allows multiplying the deflection angle by a factor of 70. The ultimate resolution of the device is 2 prad/√Hz over a measurement range of 36 µrad with a measurement band >10 kHz. The present work describes the working principle, the practical realization, and a case study on a top-level commercial angular actuator (Nano-MTA2 produced by Mad City Labs).

## 1. Introduction

High precision angular positioning is crucial in a wide range of applications, such as beam steering mechanisms and beam stabilization tasks in lithography, image stabilization, optical switches, optical disk manufacturing, and alignment of high precision mechanical stages in general.

Nowadays, precision angular actuators able to tilt objects in one or two directions with exceptional resolution are available in research institutes and on the market. Most of these actuators are piezo driven and have integrated precision metrology such as strain gauges, optical encoders, and capacitive sensors. Amongst the best performing non-commercial devices, we can mention the following ones: the piezo driven nano-angle generator coupled with grating interferometry described in [1] is capable of generating a vertical tilt with a resolution of 1 nrad over the range of about 7 mrad; [2] presents a sinebar generator capable of generating horizontal tilts with 0.5 nrad resolution and 50 nrad accuracy over a range of 80 µrad; and in [3] a similar device able to generate vertical tilts with a resolution of <1 nrad and an accuracy of 20 nrad over a range of 120 µrad is described.

The research described in [4] managed to develop a system made of a 2D tip tilting stage system based on a high stiffness sine bar structure with 10 nrad resolution over a 7 mrad range of operation for each axis. The 2D angular generator described in [5], which is driven by a piezo actuator, has a repeatability of 120 nrad and an output resolution of about 10 nrad with an operating range of about 9 mrad. Top-level 2D angular actuators can also be found off the shelf. To name a few, the S-340 piezo tip/tilt platform from Physik Instrumente has a resolution of 20 nrad with a range of 2 mrad [6] and the Nano-MTA2 from Mad City Labs has a resolution of 4 nrad and a range of 2 mrad [7].

The nominal resolution of all these devices is as low as the nanoradian level, nevertheless in most demanding applications the accuracy of the movement needs to be verified and the real resolution must be calibrated by comparison with some reference instrument. Finally, in some applications the dynamic behavior must be characterized up to the kilohertz range.

In order to test such devices, high accuracy and traceable angle sensors, such as autocollimators and interferometers, are needed.

Autocollimators are optical instruments able to measure on both axes with a relatively compact and simple setup. Nevertheless, the resolution is limited to tens of nanoradian. As an example, the top level Elcomat HR by Moller–Wedel Optical has a resolution of 25 nrad with an accuracy of ±50 nrad over any 50 µrad range [8].

The most sensitive angular measurements are based on interferometric systems. The ultra-low noise angular sensor made by a Sagnac interferometer and a folded optical lever developed at Stanford University has a resolution of 1.3 prad/√Hz and an operating frequency of 2.4 kHz [9]. An even higher resolution of 400 ± 200 frad was achieved at the University of Rochester with the help of interferometric weak value amplification [10]. The Mach–Zehnder interferometer described in [11] proved to reach a resolution of the order of 0.1 nrad/√Hz over an angular range of ±1.5 mrad, but it is not suitable for high frequencies. Commercial interferometers can be equipped with angular measurement tools typically reaching an accuracy of the order of 100 nrad [12,13].

In summary, autocollimators are able to measure tip and tilt with good accuracy but do not achieve the nanoradian resolution, while systems based on interferometry has the disadvantage of being quite complex and generally measure only an axis at a time; furthermore, they are generally rather slow.

The instrument described in this paper has been designed and built with the purpose of characterizing 2D angular actuators with a resolution better than 1 nrad/√Hz in the band from 0.1 Hz to 10 kHz on two orthogonal axes. It is traceable to the SI unit through comparison with a calibrated autocollimator. The paper describes the working principle, the practical realization, and a case study on a top-level commercial angular actuator.

## 2. Working Principle

The measurement principle combines the effect of the optical lever (which converts a rotation of a mirror into a translation of a laser beam) and the reflection law (the rotation of the reflected beam is twice the rotation of the mirror) in a multiple reflection set-up. The principle of the angle amplification (AA) was previously used as a high-resolution autocollimator and described in [14]. A laser beam is sent towards a pair of quasi-parallel mirrors (see Figure 1) and after *N* reflections exit from the same direction (solid line). Mirror A is glued to the tilting device to be measured while mirror B is kept fixed. When mirror A is rotated, the exiting beam is rotated and translated proportionally to both the distance between the two mirrors and *N* (dashed line). By using high reflectivity mirrors and proper angles, *N* can be around 70 or more, leading to an enormous gain. A position sensitive detector measures the displacement of the output laser beam and converts it into an electric signal. The sensitivity of the device is better than 10^−11^ rad/√Hz.

## 3. Experimental Set-Up and Case Study

In Figure 2, the experimental set-up is depicted and the key elements are highlighted. The laser beam is provided with a fiber coupled stabilized He–Ne laser (SIOS SL 02/1) ending on a fiber collimator mounted on a tilter. The laser is amplitude stabilized by its own electronics. It must be noted that in principle any laser source coupled with a single mode fiber can be used for the purpose. The collimated beam, about 1 mm wide (1/e at the lens), is sent to a folding mirror mounted on a high stability mirror mount. The combination of the collimator and the mirror mounts allows us to precisely align the beam to enter in the two-mirror assembly. The beam is reflected between the two high reflectivity mirrors and finally exits from the same side at a slightly lower height impinging on a second folding mirror that eventually sends the beam on a 2D position sensitive detector (PSD, model DLS-2 from UDT). A custom-made low noise electronics provides x and y output voltage and normalization with respect to the average power in order to reduce the effects of the residual amplitude noise of the laser. A polarizer is used to adjust the optical power on the PSD. The fixed “reference” mirror is mounted on a tip and tilt optical mount driven by two motorized screws (Picomotor, New Focus, USA).

The motorized screws are used to adjust the angle between the mirrors before the measurement set and are switched off during the measurement. The moving “measurement” mirror is fixed with epoxy glue to the device under test (DUT). Both the reference and the measurement mirrors are made of Clearceram^®^, have λ/10 flatness, and are coated with a multilayer dielectric coating for a reflectivity better than 99.8% at 633 nm. The two mirrors have a reflecting surface of 70 × 20 mm^2^ and are 10 mm thick. The distance between the mirrors is about 10 mm.

In the case presented here, we have tested the piezo driven nano-actuator Nano-MTA2 produced by Mad City Labs (Madison, WI, USA) [7] capable of 2 mrad p.p. full scale on two orthogonal axes. The MTA-2 is equipped with integrated piezoresistive PicoQ sensors with a nominal resolution of 4 nrad and a bandwidth of 400 Hz. The output of the piezoresistive sensor is used for the closed loop control of the MTA; furthermore, the same output is made available for the user in order to check the “real” angular displacement of the actuator. The MTA electronics allows to drive the control loop of the actuator with an external analog signal.

The experiment was realized in the INRIM’s “angle measurement” laboratory located underground in a thermally stabilized environment at 20.0 ± 0.1 °C.

## 4. Results and Discussion

We performed two kinds of measurements: the first one to characterize the noise of the AA and of the DUT in the frequency domain; the second one to characterize the resolution of the DUT in the time domain. All the measurements were carried out for both vertical and horizontal axes of the DUT. We used the horizontal axis to evaluate the spectral noises in Figure 3 and Figure 4, while we used the vertical axis to evaluate the noise of the piezoresistive sensor in Figure 5. The two axes had almost identical performances.

The whole experimental set-up was mounted on a stable granite table (passively isolated) and enclosed in a wooden box to isolate from acoustic noises and air turbulences. The signals coming from the PSD were amplified and sent to a 16 bit 100 kS/s A/D converter and processed by a LabView^®^ based software to calculate the noise spectral density. The results of the noise tests are summarized in Figure 3. The green curve represents the noise limit due to the laser noise, the detector noise, and the electronics noise. It was obtained by sending the laser beam on the detector after a single reflection in order to include all noise sources besides mechanical ones. The blue curve is the noise spectral density of the sensor of the AA when the DUT is switched off. The noise was limited by the environmental disturbance (mainly acoustic and seismic vibrations) coupled with the mechanical structure. The dashed red line indicates the 1 nrad/√Hz level for reference. The performance could be further improved by a better acoustic and vibration insulation, although the environment was already very quiet. Finally, the DUT was switched on and the noise curve is plotted in red. The difference between the blue and the red curves shows that when switched on, the active control system introduces a mechanical noise that cannot be estimated without an independent measurement. Finally, at frequencies above 3 kHz the angular noise of the device reached the electronics noise floor of the AA which is about 2 prad/√Hz.

The results of the time domain resolution tests are summarized in Figure 4. The actuator was driven by a square angular signal having nominal amplitude of 25 nrad p.p. at 1 Hz. Curves a and b are the output of the Nano-MTA2 sensor low pass filtered respectively at 30 Hz and 1 kHz. Curve c is the output of the AA low pass filtered at 1 kHz showing the “real” behavior of the DUT. It is evident that the output of the integrated piezoresistive sensor of the actuator underestimates the real noise of the sensor itself at low frequencies (curve a), since the measurement was performed “in the control loop.” In contrast, at high frequencies, the noise of the sensor is higher than the real noise of the actuator (curve b) which is hence overestimated. In any case, this experiment demonstrates the exceptionally good resolution of the actuator under test.

Figure 5 shows the noise of the MTA PicoQ sensor recorded when the system is at rest. This noise has the effect to limit the effective resolution of the device and, as shown in Figure 4, to overestimate the “real” angular noise of the MTA for frequencies higher than 10 Hz and to underestimate the noise for frequencies lower than 10 Hz. Considering that the full scale of the sensor is 2 mrad, the exceptional dynamic range of the sensor, exceeding 7 orders of magnitude, is self-evident.

The resolution of the angle amplifier (AA) in the present setup was limited by acousto-mechanic disturbances coupling with the mechanical resonances of the actuator, mainly present in the range between a few Hz up to 1 kHz. A wooden cover was used to reduce acoustic noise at high frequencies. Further noise reduction could be obtained by better damping of the floor vibration. Figure 6 shows the spectrum of the noise measured nearby the AA with and without the wooden cover. It is easy to find the peak at 10 Hz and the noise between 10 to 1000 Hz in the blue spectrum of Figure 3. The ultimate limit of the AA is the white noise of the electronics, which has a floor at 2 prad/√Hz.

## 5. Calibration of the AA

In order to convert the voltage generated by the PSD electronics into an angle, a calibration was needed. Although in principle it is possible to model the shift of the beam caused by the rotation of the mirror by means of basic ray tracing analysis, it was much easier and reliable to make use of a direct comparison with a reference standard. Therefore, we performed a calibration of the DUT on the full scale to check for its scale factor and linearity. The calibration was undertaken by direct comparison with a reference autocollimator (ELCOMAT HR by Moeller–Wedel, Wedel, Germany), in turn calibrated against the INRIM angle standard [15].

The autocollimator (AC) was placed at the back side of the AA pointing at an auxiliary mirror glued on top of the mirror fixed to the MTA. The MTA was driven in steps of known nominal value and the output of the AC was recorded together with the output of the AA electronics. The slope in the interval was calculated for both axes (being about 1.8 µrad/V) and was used to convert the output of the ADC board directly into radian units. The uncertainty of the conversion factor was calculated taking into account the uncertainty and the resolution of the AC and the non-linearity of the PSD sensor in the area of interest. The full-scale range in the presented configuration was 36 µrad ≈ 7.4 arcsec. An uncertainty less than 0.1% was assigned to the scale factor. For angular intervals of the order of the microradian this uncertainty was dominant; while at the nanoradian scale the error sources coming from the noise were dominant and the time interval of the measurement must be taken into account.

## 6. Conclusions

We realized a simple and effective set-up to calibrate and characterize precision piezoelectric nano-angle actuators on two orthogonal axes. The method surpassed both state of the art autocollimators in resolution and bandwidth and interferometers in terms of compactness and simplicity. The noise of the device (here called AA) was limited by environmental disturbances to less than 1 nrad/√Hz for frequencies higher than 0.1 Hz and was as low as 2 prad/√Hz for frequencies higher than 3 kHz. The working range was 36 µrad and the measurement bandwidth was >10 kHz. It might be noted that the working range can be easily extended by reducing the number of reflections i.e., the gain of the AA. This can be done by simply rotating the reference mirror, without changing the setup. Obviously, the resolution and noise will be scaled down by the same factor. The device showed better performance than any measurement device or angular actuator available on the market to the knowledge of the authors. The metrological traceability was guaranteed through the use of a reference autocollimator calibrated against the INRIM angular standard. With a practical experiment we demonstrated the feasibility of the calibration and dynamic characterization of a top-level commercial device. The method allowed us to find the real performance of the DUT, demonstrating that by relying on the internal sensor of the DUT, the noise can be underestimated or overestimated depending on the frequency band.

It has to be noted that the AA can be used to characterize the actuators only where it is possible to attach a mirror of adequate size. Indeed, the measuring principle requires a high reflection surface of several centimeters, thus, although it could be reduced in scale, a finite surface area is still needed. Consequently, it cannot be used to replace autocollimators to measure very small actuators, or to characterize the local deviations from flatness. Furthermore, when compared with commercial interferometers it has the limitation of a smaller measurement range.

## Figures and Tables

**Figure 1 sensors-20-07034-f001:**
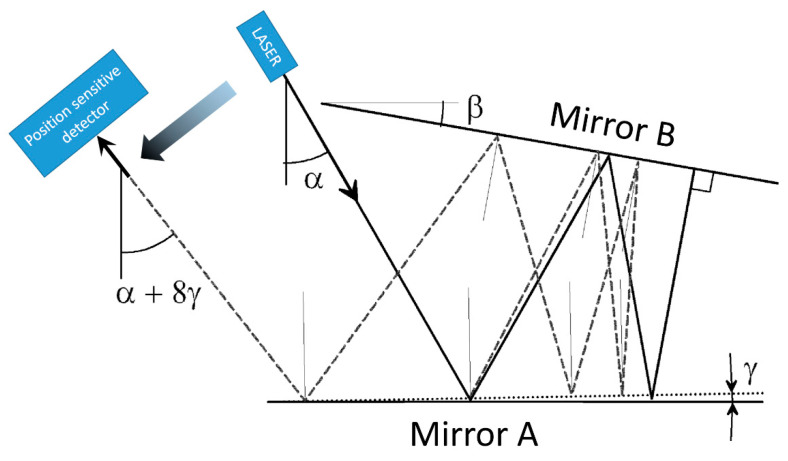
Ray tracing of the multiple reflection set-up showing the effect of the counter clockwise rotation of mirror A on the reflected beam. All angles in the figure are excessive for the sake of clarity. Incidence angle α is 30°; angle β between A and B is 10°; mirror B is rotated by γ = 1°. In the initial condition, after three reflections (α/β = 3) the laser beam impinges orthogonally on mirror B and is reflected back on its path. This auto collimation condition is close to the operative condition of the device. When mirror A is rotated by 1° counter clockwise, the beam is moved towards left and is rotated by 8 times γ.

**Figure 2 sensors-20-07034-f002:**
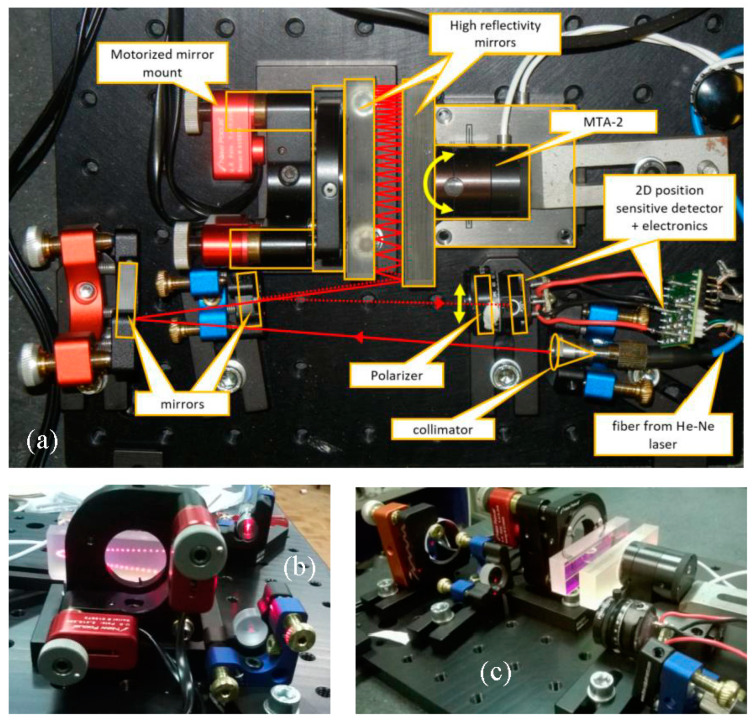
Pictures of the experimental set-up. (**a**) top view with the laser path drawn on the picture: the entering laser beam (solid red line) is sent towards the multiple reflection mirrors where the angle amplification occurs (see text for the details). The contours in orange indicate the main optical components and the yellow arrows indicate the movement of the MTA-2 and the translation of the beam; (**b**) back view of the angle amplification (AA); the multiple reflection pattern is visible at the back of the “reference” mirror. (**c**) side view of the AA; the device under test (DUT) holding the “measurement” mirror is visible at the top right.

**Figure 3 sensors-20-07034-f003:**
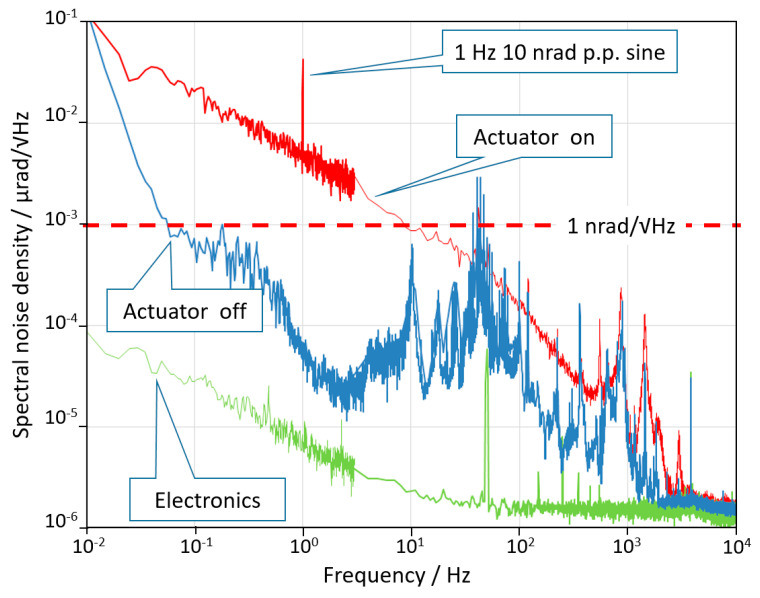
Noise spectral density curves of the output signal of the measuring device in three different conditions. The meaning of the three curves is explained in the text. *y*-axis analysis is not shown, but its behavior is very similar to that of *x*-axis.

**Figure 4 sensors-20-07034-f004:**
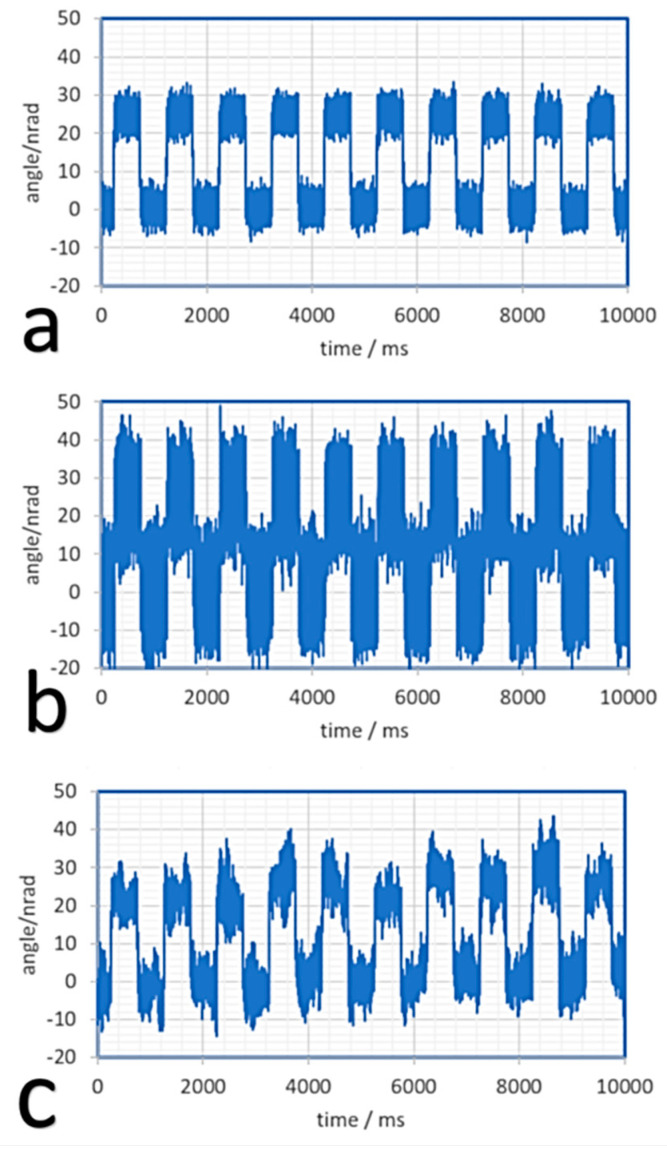
Time domain characterization of the MTA-2. The sensor is driven with a square wave having amplitude 25 nrad p.p. and 1 Hz frequency. In (**a**) the output of the MTA monitor filtered at 30 Hz; in (**b**) the same output filtered at 1 kHz; in (**c**) the output of the AA filtered at 1 kHz.

**Figure 5 sensors-20-07034-f005:**
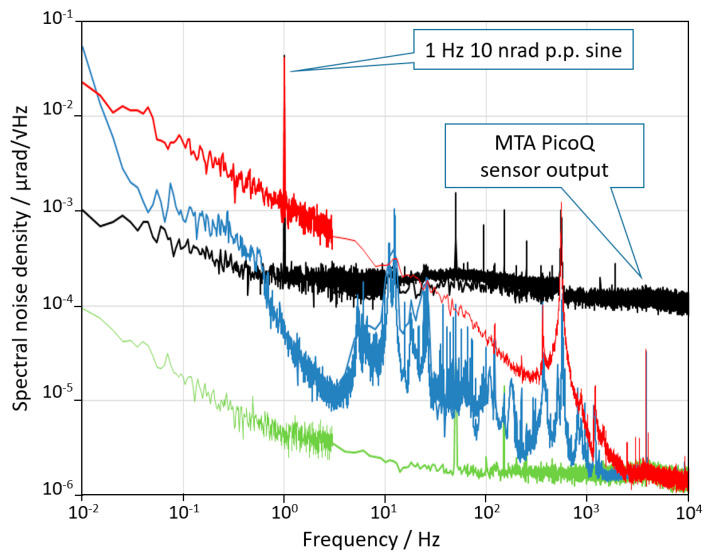
Noise of the MTA sensor: the black curve added to the noise spectral density analysis shows the noise of the MTA PicoQ sensor, which has the effect to limit the actual resolution of the device and, as shown in Figure 4, to overestimate the “real” angular noise of the MTA at high frequencies. Note that the noise spectra of the AA in this figure refer to the vertical axis; this shows the similar behavior of the two axes.

**Figure 6 sensors-20-07034-f006:**
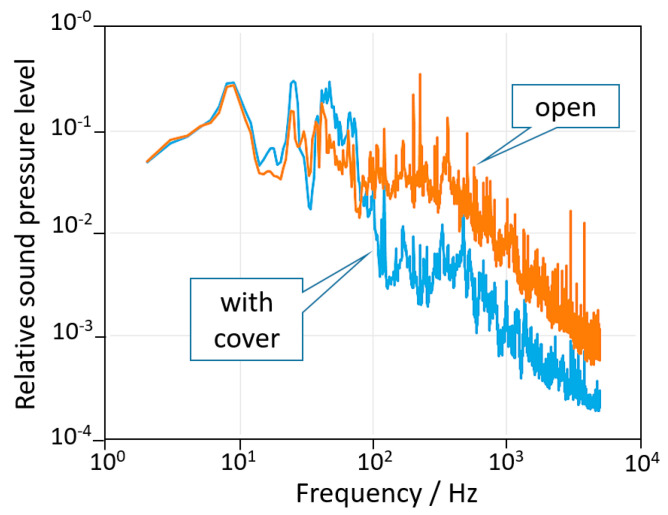
Relative sound pressure spectrum measured in proximity of the AA with and without the wooden cover. The microphone was not calibrated, therefore the vertical scale has been normalized to an arbitrary intensity level. It is evident that the cover is more effective in the acoustic region above 100 Hz and less effective in the 10–100 Hz range dominated by seismic noise. The latter could be reduced by improved damping of floor vibrations.

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
