# Peer review of "Sensor for the Characterization of 2D Angular Actuators with Picoradian Resolution and Nanoradian Accuracy with Microradian Range"

_sensors, 2020, doi:10.3390/s20247034_

Round 1
Reviewer 1 Report
• In the abstract, the reviewer did not explain the novelty of this paper very clearly.
• The references are enough, but it is better to have a summary results to make the novelty more clear.
• Figure 4 is hard to understand, what does the author want to express?
• Conclusion should be improved.
• Figure 1 is not clearly, and it should be corrected. It is hard to recognize the physical part. The label positions are randomly.
In summary, most figures are not professional.
Author Response
Author’s answers to Reviewer 1
We thank the reviewer for the useful comments, criticism and suggestions that contributed to greatly improve the paper. In the revised version, the changes are highlighted in red.
Comment: In the abstract, the reviewer did not explain the novelty of this paper very clearly.
Answer: True. We have rewritten the first sentences to highlight the fact that the characterization of high-resolution angular actuators cannot be performed with available instruments, thus requires the realization of a novel method.
Comment: The references are enough, but it is better to have a summary results to make the novelty more clear.
Answer: In fact, without numbers it is not evident the difference between the state of the art measurement instruments and the instrument presented in the paper. Consequently, the novelty cannot be appreciated. We have implemented the introduction by:
- Briefly describing the performances of the state of the art actuators cited in the references, in terms of resolution and range or other relevant parameters;
- Adding to the list two state of the art commercial actuators (one of which is used in the paper).
From this should be clear the order of magnitude of the angles to be measured if we wish to characterize the above instruments. Then we have:
- Commented the performances of the best sensors available in terms of resolution, range or other relevant parameters;
- Added two representative top level commercial instruments (interferometer and autocollimator).
- Removed from the references the paper describing applications that were a bit out of topic such as the calibration of autocollimators and the use of autocollimators for measuring flatness.
Finally, we have introduced our device. Now the meaning of the work should be clearer.
Comment: Figure 4 is hard to understand, what does the author want to express?
Answer: In fact, we realized that the meaning of the picture was clear only for the authors. The point is to demonstrate with graphs that if we rely only on the output of the driver (which is intended to show the “real” displacement of the actuator) we make important interpretation errors. With our measurement, we demonstrate that the real movement is different from what measured. To improve the picture we have:
- Reduced the number of curves from 5 to 3: two from the MTA output and one from the AA output.
- Included the three curves in their own reference frame with clear indication of vertical and horizontal scales.
- Rewritten the caption in order to explain the meaning of the three curves.
Comment: Conclusion should be improved.
Answer: We have implemented the conclusions by highlighting the advantages of the method with respect to state of the art sensors; furthermore, we have listed the limits of the AA, mainly in the number of applications. I.e. the AA is not intended to replace interferometers, nor autocollimators, but to fill a measurement gap.
Comment: Figure 1 is not clearly, and it should be corrected. It is hard to recognize the physical part. The label positions are randomly.
Answer: We guess that the comment refers to Figure 2. Indeed the figure was awfully misaligned. We have adjusted the picture by:
- Correcting the position of the contours
- Changing some label and the position of the same
- Changing the colours of the lines on the picture
- Adding some more details in the caption
Comment: In summary, most figures are not professional
Answer: We believe that thanks to the reviewer comment the figures are now more readable.
We believe that changing Figures 2 and 4 and working on the captions now the paper appear more professional.
Reviewer 2 Report
Manuscript ID: EPJP-D-20-00396
Title: Sensor for the characterization of 2D angular actuators with picoradian resolution and nanoradian accuracy on the microradian range
Authors: Marco Pisani, Milena Astrua and Srijith Thirumalaj.
Anamnesis:
The letter reports on the realization of a calibration set-up for the analysis of precision of nano-angular actuators. The manuscript begins with a short introduction on the issue connected to this kind of measurements giving a glance on the state of art in the field. Then, they describe the experimental set-up and discuss methodology and measurement results in details.
Comments:
I found the paper very well written. The experimental description is very detailed and the reader can easily follows all the measurement procedure. There are just few non-mandatory remarks:
- I would appreciate just one plot more for proving the effectiveness of the method also on the vertical axis.
- Improving Caption of Fig.4 would be helpful. Reading the details from the text is not so comfortable
- The discussion on the acoustic peak around 10 Hz clearly visible in Fig. 6 (lines 169-171) can be moved at an earlier point so to better explain this peak already evident in Fig. 3. One possible choice is to move Fig.6 as an inset in Fig. 3.
- Line 114 the second “horizontal” is probably “vertical”
- Line 124 “mechanical” in the round brackets could be “acoustic” so to avoid repetition of the same word in the same sentence
- Line 153-154 there is an extra “full scale”
Final Recommendation
I believe that the paper can be published nearly as it is.
Author Response
Author’s answers to Reviewer 2
We thank the reviewer for the useful comments, criticism and suggestions that contributed to greatly improve the paper. In the revised version, the changes are highlighted in red.
Anamnesis:
The letter reports on the realization of a calibration set-up for the analysis of precision of nano-angular actuators. The manuscript begins with a short introduction on the issue connected to this kind of measurements giving a glance on the state of art in the field. Then, they describe the experimental set-up and discuss methodology and measurement results in details.
Comments:
I found the paper very well written. The experimental description is very detailed and the reader can easily follows all the measurement procedure. There are just few non-mandatory remarks:
- I would appreciate just one plot more for proving the effectiveness of the method also on the vertical axis.
Answer: In fact, in figure 5, where we show the noise of the piezoresistive sensor, we have used as a reference the spectra of the vertical axis. That was not mentioned in the previous version. Now we have clarified it in the second sentence of chapter 4 and in the caption of figure 5. So, we believe it is easy to see the similar behaviour of the two axes.
- Improving Caption of Fig.4 would be helpful. Reading the details from the text is not so comfortable
Answer: In fact, the picture was not clear at all. We have changed the picture, simplified it, added the scales and improved the caption. We believe that the picture is now more clear.
- The discussion on the acoustic peak around 10 Hz clearly visible in Fig. 6 (lines 169-171) can be moved at an earlier point so to better explain this peak already evident in Fig. 3. One possible choice is to move Fig.6 as an inset in Fig. 3.
Answer: We agree that this solution could have improved the reading, nevertheless, this would have changed too much the figure 3, so we preferred to leave figure 6 in its place. We have improved the caption of the same.
- Line 114 the second “horizontal” is probably “vertical”.
Answer: We have changed the whole sentence as a consequence of point 1
- Line 124 “mechanical” in the round brackets could be “acoustic” so to avoid repetition of the same word in the same sentence
Answer: We have changed the text in brackets according to the suggestion
- Line 153-154 there is an extra “full scale”
Answer: Thank you for noticing: corrected
Reviewer 3 Report
This paper presents a method for measurement of small angular displacement with the multiple-reflection setup. The feasibility of the proposed setup has successfully been demonstrated through experiments, and the reviewer believes the letter paper can be accepted for publication.
Meanwhile, the authors are expected to address the following comments from the reviewer.
1) Although the meaning of the multiple plots in Fig. 4 is stated in the sentences, the authors are expected to tell the difference in the figure itself for the sake of readability.
2) Detailed information on the resolution has been well reported in the manuscript including the abstract and the conclusion. Meanwhile, it seems that the information on the achievable measuring range is not clearly indicated. The authors are expected to mention the measuring range to be achieved by the developed system.
3) In addition, the authors are expected to explain how to apply the developed system to practical measuring applications. One of the advantages of the conventional laser autocollimator is its easiness of the setup in practical applications. Regarding the principle of the developed setup, it seems to be a little bit difficult to apply the system in industrial applications. The authors are encouraged to address the above mentioned issue, while discussing the pros and cons of the proposed method.
Author Response
Author’s answers to Reviewer 3
We thank the reviewer for the useful comments, criticism and suggestions that contributed to greatly improve the paper. In the revised version, the changes are highlighted in red.
This paper presents a method for measurement of small angular displacement with the multiple-reflection setup. The feasibility of the proposed setup has successfully been demonstrated through experiments, and the reviewer believes the letter paper can be accepted for publication.
Meanwhile, the authors are expected to address the following comments from the reviewer.
- Although the meaning of the multiple plots in Fig. 4 is stated in the sentences, the authors are expected to tell the difference in the figure itself for the sake of readability.
Answer: In fact, the picture was not clear at all. We have changed the picture, simplified it, added the scales and improved the caption. We believe that the picture is now more clear.
- Detailed information on the resolution has been well reported in the manuscript including the abstract and the conclusion. Meanwhile, it seems that the information on the achievable measuring range is not clearly indicated. The authors are expected to mention the measuring range to be achieved by the developed system.
Answer: In fact, the measuring range is mentioned only in the abstract and in chapter 5. As the reviewer pointed out, this is a key parameter that must be put in evidence. We have added the full range (36 microrad) also in the conclusions. Furthermore, in the conclusions, we have added a sentence to explain that the measuring range can be extended by reducing the number of reflections of the AA, consequently worsening the resolution.
- In addition, the authors are expected to explain how to apply the developed system to practical measuring applications. One of the advantages of the conventional laser autocollimator is its easiness of the setup in practical applications. Regarding the principle of the developed setup, it seems to be a little bit difficult to apply the system in industrial applications. The authors are encouraged to address the above mentioned issue, while discussing the pros and cons of the proposed method
Answer: We thank the reviewer for the constructive comments. In fact, the introduction mentions autocollimators citing amongst the others their use for measuring flatness. It is true that the presented method is not applicable to this purpose. Indeed, the AA is not intended to replace interferometers, nor autocollimators, but to fill a measurement gap. For this reason, we have rewritten the introduction removing the reference to the applications of the autocollimators that cannot be applied to our device. Furthermore, we have mentioned this limitation and others in the conclusions.
Reviewer 4 Report
Dear authors!
The peer-reviewed article presents a very useful and ingenious solution of the hard problem, which researchers meet when it is necessary to measure angular deviations with extremely high accuracy. The principle of the developed method is enough simple, but its realization may face insurmountable difficulties in solving specific practical problems. The authors convincingly presented the limitations inherent in the method, primarily due to the influence of external noise and measuring equipment noise on its sensitivity and resolution. Moreover, as follows from the presented materials, the method is applicable only in cases where the analyzed surface has a very high reflectivity. These limitations do not diminish the importance and originality of the proposed method, but make it desirable to correctly outline the range of problems available for solving by the developed method. For example, it is not clear how this method can be used in the study of the flatness of the surface of the plates and profilometry of the surface, which is stated in the Introduction. Also, environmental conditions will affect the effectiveness of the method, narrowing the scope of its application. Some recommendations on the selection of the optimal frequency range, which can be made based on the results of experiments, seem desirable.
The article is very well organized, the presented research technique is described in detail, the conclusions are supported by the results.
Author Response
Author’s answers to Reviewer 4
We thank the reviewer for the useful comments, criticism and suggestions that contributed to greatly improve the paper. In the revised version, the changes are highlighted in red.
Comment: The peer-reviewed article presents a very useful and ingenious solution of the hard problem, which researchers meet when it is necessary to measure angular deviations with extremely high accuracy. The principle of the developed method is enough simple, but its realization may face insurmountable difficulties in solving specific practical problems. The authors convincingly presented the limitations inherent in the method, primarily due to the influence of external noise and measuring equipment noise on its sensitivity and resolution. Moreover, as follows from the presented materials, the method is applicable only in cases where the analyzed surface has a very high reflectivity. These limitations do not diminish the importance and originality of the proposed method, but make it desirable to correctly outline the range of problems available for solving by the developed method. For example, it is not clear how this method can be used in the study of the flatness of the surface of the plates and profilometry of the surface, which is stated in the Introduction. Also, environmental conditions will affect the effectiveness of the method, narrowing the scope of its application. Some recommendations on the selection of the optimal frequency range, which can be made based on the results of experiments, seem desirable.
Answer: We thank the reviewer for the constructive comments. In fact, the introduction mentions autocollimators citing amongst the others their use for measuring flatness. It is true that the presented method is not applicable to this purpose. For this reason, we have removed the reference to this application from the introduction and from the references. Furthermore, we have mentioned this limitation and others in the conclusions.
Round 2
Reviewer 1 Report
I think this paper needs a major revision or reject, and figure 2 did not show clearly. There is no unit in the vertical coordinates of figure 6. This paper needs more work to show the ideas.
Author Response
Dear Reviewer, we have changed the scales to all figures and solved the issue of the resolution of the pictures that was dure to a inaccurate pdf conversion. We sincerely hope that this version is satisfying for you.